

# Mobile and stationary sources of air pollutants in the Amazon rainforest: a numerical study with WRF-Chem model

Sameh A. Abou Rafee[1], Leila D. Martins[1], Ana B. Kawashima[1], Daniela S. Almeida[1], Marcos V.B. Morais[1], Rita V.A. Souza[2], Maria B.L. Oliveira[2], Rodrigo A.F. Souza[2], Adan S.S. Medeiros[2], Viviana Urbina[3], Edmilson D. Freitas[3], Scot T. Martin[4], Jorge A. Martins[1]

[1]Federal University of Technology – Parana, Londrina, Brazil
[2]Amazonas State University – Amazonas, Manaus, Brazil
[3]Department of Atmospheric Sciences, University of São Paulo, São Paulo, Brazil
[4]Harvard University, Cambridge, Massachusetts, USA

*Correspondence to*: Sameh A. Abou Rafee (samehabou@gmail.com), Jorge A. Martins (jmartins@utfpr.edu.bt)

**Abstract.** This paper evaluates the impact of the emissions from mobile and stationary sources in the Amazon rainforest by using the Weather Research and Forecasting with Chemistry (WRF-Chem) model. The analyzed air pollutants were CO, $NO_x$, $SO_2$, $O_3$, $PM_{2.5}$, $PM_{10}$ and VOCs. Five scenarios were defined in order to evaluate the emissions by biogenic, mobile and stationary sources, as well as future scenarios. Results show that the stationary sources explain the highest concentrations for all air pollutants evaluated, except for CO, for which the mobile sources are predominant ones. The futuristic scenario, which is twice the current emissions from mobile and stationary sources, has shown an increase in the range of 3 to 62% in average concentrations and 45 to 109% in peak concentrations depending on the pollutant. In addition, the spatial distributions of the scenarios has shown that the air pollutions plume from the city of Manaus is predominantly west and southwest, and it can reach hundreds of kilometers in length.

## 1 Introduction

The impacts of anthropogenic emissions from urban areas on the environment and human health have been covered in several scientific research through different methodologies (Kampa & Castanas, 2008; Martins et al., 2010). These analyses have shown that the consequences of such emissions go beyond their initial source and can reach even farther during long periods. In general, most recent studies focus mainly on the aftermath of the anthropogenic emissions from the megacities in the developing world (e.g., Zhu et al., 2013). This has happened because such cities have increased their energy demand, which has been high throughout recent decades (Lutz et al., 2001). For instance, energy consumption in China has increased more than 300 million tons of coal equivalent (MtCE). Most of this energy is produced from burning fossil fuels, mainly coal (Crompton and Wu, 2005). As a consequence, the atmosphere in these regions has experienced a great increase not only of particles but also of gases such as carbon monoxide (CO), sulfur dioxide ($SO_2$), oxide nitrogen ($NO_X$), and other secondary compounds such as ozone ($O_3$). Understanding the impact of each pollutant on the environment and on public health has strategic importance to the creation of new public policies and to the development of brand-new technologies, which guarantee the improvement of air quality.

Determining what is the real contribution of the anthropogenic air pollutants to the environment and to human health is not a straightforward task. Its difficulty lies in separately studying the emissions from natural sources and human sources and the complex combination of the latter. Although urban areas can have an identity



associated to their economic profile, the transportation and the combination of atmospheric pollutant effects hamper the understanding of the role of each emission species in the studied region. In this context, atmospheric

models with an explicit treatment of the physical and chemical processes become an indispensable tool to the studies of the urban pollution impact. Several numerical studies can be found with special focus on different aspects of the pollution impact. For instance, pollution episodes (Jiang et al., 2012; Michael et al., 2013; Kuik et al., 2015), regional and long distance transportation (Tie et al., 2007; Guo et al., 2009; Lin et al., 2010), secondary formation of gases and particles (Yerramile et al., 2010; Jiang et al., 2012; Lowe et al., 2015), and the effects on

land use and land cover changes (Capucim et al., 2015; Rafee et al., 2015). However, such studies are not capable of investigating the impact of isolated urban plumes.

The pronounced growth of urban areas in the last decades, mainly due to the accelerated population growth rate and the migratory flow to cities, resulted in few areas in the world where an isolated urban area would interact with an environment of natural and homogeneous characteristics. The city of Manaus is one of the unique

scenarios in the world where an isolated urban area is in the center of a vast tropical forest area, the Amazon rainforest (Martin et al., 2016a). Thus, the region is a valuable laboratory for studying the impact of anthropogenic emissions of air pollutants. Although, there are a number of studies involving the measurements of atmospheric pollutants in the Amazon region (e.g., Kuhn et al., 2010; Trebs et al., 2012; Baars et al., 2012; Artaxo et al., 2013), most of the contribution was published very recently with the results of the GoAmazon project (Martin et al.,

2016a; Martin et al., 2016b; Alves et al., 2016; Pöhlker et al., 2016; Sá et al., 2016; Kourtchev et al., 2016; Hu et al., 2016; Bateman et al., 2016; Cecchini et al., 2016; Jardine et al., 2016). In terms of numerical studies, only a few studies focusing the role of the anthropogenic emission sources in urban area are available (Freitas et al., 2006; Andreae et al., 2012; Beck et al., 2013; Bela et al., 2014). Therefore, such studies do not address the participation of each type of emission source on air pollutants. This type of evaluation can only be performed with

the use of atmospheric modeling tools, which require the preparation of an inventory for mobile and stationary sources.

Given the importance of anthropogenic emissions in the Amazon region, this paper reports a numerical study evaluating the impact of the urban pollution plume on the preserved forest region considering individual contributions from the main mobile and stationary sources in Manaus. In addition, a companion study of Medeiros

et al., 2016 focus how the changing energy matrix for power production affects air quality in Manaus region. Both studies were conducted through the Weather Research and Forecasting with Chemistry (WRF-Chem) model, where the simulations for diverse scenarios were performed according to the current conditions of the region. The numerical experiments performed, in this work, addressed the following issues:

- Participation of mobile and stationary sources in the concentration of trace gases and particles for the city of
Manaus and adjacent areas directly influenced by it;
- Preferential direction and the distance of the urban plume impact from Manaus in the Amazon region;
- Impact on air quality due to the likely future increase in anthropogenic emissions from mobile and stationary sources.





## 2 Methodology

### 2.1 Area and Period of Study

The scope of the study comprises the urban area of Manaus and its surroundings, with a total area of 230,560 km$^2$ (Figure 1). Manaus is located in northern Brazil, at latitude $\lambda = 3°06'07''$ and longitude $\varphi=60°01'30''$ with an urban area of 377 km$^2$, presented in the center of the grid below. Currently, Manaus has an estimated population of 2 million people representing more than 50% of the population in the state of Amazonas, with 99.49% of its population living in urban areas (IBGE, 2014).

The simulation period encompasses the dates of March 16 – 18, 2014, and this period represents the rainy season of the region (Fisch et al, 1998). Due to the scarce availability of air quality data, the choice of the days for simulations were made according to that availability, necessary to evaluate the performance of the model with the observed ground-based data. Four measurement sites were available in different locations in the study area, as presented in Figure 1: T1, located within the city of Manaus, at the National Institute for Amazonian (INPA); T3, located in the north of Manacapuru, approximately 100 km from Manaus, at stations of the project Green Ocean Amazon (GoAmazon, 2014); Colégio Militar and Federal Institute of Amazonas (IFAM), located in the city of Manaus and the fourth is associated to the Project REMCLAM Network of Climate Change Amazon from the University of the State of Amazonas (UEA). According to data from stations located in the city of Manaus, there was no registration of rainfall within the given period, whereas Manacapuru site recorded 7.2 mm rainfall. Regarding the predominant wind direction, the ground-based data observed a prevalence of winds from the north, northeast and east. In terms of wind direction climatology, the prevailing northeasterly/westerly winds blow all year round, 10°-60° azimuth (Nov. to Mar.) and 90°-130° azimuth (May to Sept.) (Araújo et al., 2002). Another important factor when choosing the period of study was the low incidence of biomass burning wildfires in the rainy season. Fire outbreaks in the Amazon region are predominant during the dry season and can therefore affect the concentration of particulate matter and trace gases (Andreae et al., 2001; Martins & Silva Dias, 2009). Since the purpose is only to analyze the impact of anthropogenic emissions attributed to fossil fuels, periods with the presence of regional wildfires would not be ideal.

### 2.2 Model Description

In this study, the coupled WRF/Chem (*Weather Research and Forecasting with Chemistry;* Grell et al., 2005) model, version 3.2 was applied. The model is an online system that predicts meteorological and chemical states, simultaneously. WRF/Chem or its previous versions have been applied in a number of studies worldwide (e.g., Grell et al., 2005; Wang et al., 2009; Wang et al., 2010; Tuccella et al., 2012; Vara-Vela et al. 2015). Physical parameterizations used in this study are shown on Table 1.

For the biogenic emission, the module used was based on descriptions by Guenther et al. (1993 and 1994), Simpson et al. (1995) and Schoenemeyer et al. (2007). This module deals with isoprene, monoterpenes, volatile organic compounds (VOCs) emissions and emissions of nitrogen from the soil. Biogenic emissions were calculated using the categories of land use and soil occupation available in the model in which emission rates are estimated from the temperature and photosynthetic active radiation, which is the fraction of solar radiation comprised in the range of the visible spectrum available (0.4 to 0.7 μm) for further photosynthesis process. The aerosol parameterization used was based on Modal Aerosol Dynamics model for Europe - MADE (Ackermann et





al., 1998) developed from the Regional Particulate Model - RPM (Binkowski and Shandar, 1995), embedded with the representation of organic aerosol side, SORGAM (Secondary Organic Aerosol Model developed by Shel et al., 2001).

For the gas phase chemistry, the chemical mechanism used was the Regional Acid Deposition Model Version 2 (RADM2, Chang et al., 1989), originally developed by Stockwell et al. (1990). RADM2 is widely used in atmospheric models to predict concentrations of oxidants and other pollutants and contains 158 chemical reactions, of which 21 are photolysis.

    The simulation domain was configured with a 3-km horizontal spacing grid, with 190 grid points in x and 136

grid points in y, centered in the city of Manaus (3° 4 '12"S; 59° 59' 24"W). In the vertical grid, 35 levels were defined, with the top at 50 hPa, corresponding to about 20 km in height. Analysis data from the Global Model Data Assimilation System (GDAS), with a horizontal spacing grid of 1° and 26 vertical levels were used for initial and boundary conditions of the meteorological variables. For chemical variables, the initial and boundary conditions of simulations consist of idealized, northern hemispheric, mid-latitude, clean environmental conditions

as described in Liu et al. (1996) and applied in studies, such as those conducted by Grell et al. (2005), Wang et al. (2009), Wang et al. (2010), Tuccella et al. (2012), Vara-Vela et al. (2015). On the other hand, there are studies where the chemical compound profiles used as initial and boundary conditions are based on anthropogenic emission inventories, including CO, $NO_x$, $SO_2$, speciated VOC, black carbon (BC), organic carbon (OC), and particulate material (PM), at $0.5° \times 0.5°$ (e.g., Zhang et al., 2009) or higher resolution (e.g., Tie et al., 2010). Initial

and boundary conditions for chemical species have also been extracted from the output of global chemical transport models, such as MOZART - Model for OZone And Related chemical Tracers (Hu et al., 2010). However, it is important to evaluate the global model results before deciding which global model to use in order to provide the chemical initial and boundary conditions, considering that there are many uncertainties among the global results.

Different time to avoid a spin-up effect are suggested in literature, with values ranging from 12 (Tuccella et al., 2012; Carvalho et al., 2015), 24 (Wang et al., 2009), 36 (Hu et al., 2010), to 48 (Tie et al., 2010) hours of simulation as a model spin-up. There are also studies considering a few days as a model spin-up (e.g., Wang et al., 2010). In this study, a 24-hour spin-up model was used. For land use and soil occupation, spatial data from the Moderate-resolution Imaging Spectroradiometer file (MODIS) – 2005 was utilized, with 500-meter spacing

grids (Schneider et al., 2009).

### 2.3 Anthropogenic Emission Inventories

### 2.3.1 Vehicle Emissions

Emissions of all classes of light-duty and heavy-duty vehicles in the study region were taken into consideration for mobile sources. In order to calculate the emissions of air pollutants, information was collected based on the

estimate of the number and type of vehicles, emission factors and average vehicle-use intensity. The calculations included individual contributions of five types/fuel combinations of vehicles (Table 2), considering the data estimated from the Brazilian National Department of Traffic (DENATRAN, 2014) for the city of Manaus. The urban area of Manaus concentrates 83.22% of vehicles in the state of Amazonas, corresponding to over 600 thousand vehicles in the current fleet. Therefore, the fraction of the vehicle fleet according to type and fuel

consumption for the entire grid study was considered based on data from Manaus. The fractions corresponding to





light-duty vehicles using gasohol, ethanol and flex fuel represent 22.81%, 2.49% and 42.29%, respectively. Heavy-duty diesel-powered vehicles represent 8.8%, and motorcycles using gasohol represent 23.61%.

The emission factors for different vehicle types/fuel assumed in the calculations were based on experiments conducted inside the road traffic tunnels in the city of São Paulo (Martins et al., 2006; Sanchez-Ccoyllo et al., 2009; Brito et al., 2013). The values adopted for CO, $NO_x$, $SO_2$, particulate matter (PM) as well as the distribution of VOCs emission fractions related to evaporative, liquid and exhaust emissions for vehicles burning gasohol, ethanol and diesel. These values correspond to those that have been recently used in the study by Andrade et al. (2015) and are listed in Table 2. Based on these emission factors, diurnal profile using hourly variations for emissions of trace gases and PM were used in the model (Martins et al., 2006; Andrade et al., 2015). The distribution of the fractionation of emission of fine particulate matter, as well as the chemical characterization, were obtained based on several studies in the city of São Paulo for the measurements of the concentration mass and number of particulate matter (Ynoue and Andrade, 2004; Miranda and Andrade, 2005; Albuquerque et al., 2011), and are used in the investigations by Andrade et al. (2015) and Vara-Vela et al. (2015).

In order to calculate the vehicle-use intensity, reference estimates from the first Brazilian National Emissions Inventory of Road Motor Vehicles (MMA, 2011), DENATRAN and the Brazilian National Agency of Petroleum (ANP, 2014a) were used. The intensities for light-duty vehicles, heavy-duty vehicles and motorcycles are 21.3, 128.4 and 70.6 kilometer per day, respectively.

The spatial distribution of the number of vehicles in each grid points was based on nighttime lights from Defense Meteorological Satellite Program-Operational Linescan System (http://ngdc.noaa.gov/eog/dmsp/downloadV4composites.html), considering the size of urban occupation. Martins et al. (2008) have described this approach, which has performed a luminance calibration according to the population density and number of vehicles, presenting a good correlation between the parameters.

### 2.3.2 Stationary Source Emissions

The emission of thermal power plants (TPPs) of different types of fuel burning have been considered for the inventory of stationary sources in the studied region and the refinery located within the urban area of Manaus. It is noteworthy that the contributions of the industries located within the urban area have little significance in the region, due to the production concentrated mainly in transport and communication areas (Manaus, 2002). In this case, the contributions on anthropogenic emissions occur indirectly by the high consumption of electricity supplied by the TPPs. The emission of the pollutants per type of TPP in each grid point of study has been calculated according to the estimates of emission factors, fuel consumption and power generation, using the following equation (1):

$$E_{p(i,j)} = \sum_{k=1}^{N} EF_P \times FC \times PG_{(i,j,k)}, \qquad (1)$$

where $E_{p(i,j)}$ represents the emission of pollutant P at each grid point $(i, j)$ in grams per hour (g $h^{-1}$), $EF_P$ is the emission factor of pollutant P in grams per liter (g $L^{-1}$), FC is the fuel consumption in liters per kilowatt-hour (L $kWh^{-1}$), and $PG_{(i,j,k)}$ is the power generation of TPP (k) at each grid point $(i, j)$ in kilowatt (kW).

According to the Generation Database (BIG) of the National Electric Energy Agency (ANEEL, 2014), Brazil has 1,890 TPPs in operation with an installed capacity of about 37.8 GW. Based on the information contained in BIG, a spatial distribution of TPPs has been performed at each grid point of the area of study, with 59 diesel, 6





fuel oil and 8 natural gas power plants (Figure 2) found in the grid, generating a power of 1,851 MW,

corresponding to 96.76% of all generations in the state of Amazonas. The other TPPs located in state are very small.

Due to several types of technologies used in the burning fuels of TPPs, the emission factor admitted in the calculation was the intermediate values between the lower and upper limits adopted by the US Environmental Protection Agency (EPA, 1998 and 2010). The emission factor is described in Table 3, and for comparative

purposes, the values adopted in the inventory of São Paulo Environmental Protection Agency (CETESB, 2009) have also been listed. The distribution of PM has been carried based on the fractionation designed for vehicle emissions. In addition, the speciation of VOC emissions have been performed for each fuel type TPP estimated by EPA (1998; 2010). Regarding the average fuel consumption for each type of TPP, the average value of fuel to produce one kWh of power generation has been adopted, as described in the annual report of isolated operation

systems for the Northern region (ELETROBRAS, 2013). For fuel oil, diesel and natural gas, the values of 0.27 L kWh$^{-1}$, 0.29 kg kWh$^{-1}$, 0.28 m$^3$ kWh$^{-1}$, respectively, have been considered.

Some approximations that have been admitted to obtain the emission factor values of Table 3 were:

- Fuel with 1% sulfur content was admitted to calculate the emission factor of $SO_2$;
- Total organic compounds (TOCs) include VOCs, semi-volatile organic compounds, and condensable organic

compounds;

- For $NO_x$ emissions by fuel oil power plants, the minimum value was attributed to the lowest value available for vapor generation above 50 t/h and the maximum value was assigned to the greater value available for plants below 50 t/h;
- For the emission factors attributed to $SO_2$ and PM, the minimum and maximum values were defined based

on the grade of fuel burned. In this case, the combustion of lighter distillate oils results in significantly lower PM formation than those from the combustion of heavier residual oils. According to the values proposed by EPA, burning N° 4 or N° 5 oil usually produces less $SO_2$ and PM than the burning of heavier N°. 6 oil.

Brazil has 17 petroleum refining units with a production of about 769 million m$^3$ of petroleum per year (ANP, 2014b), with only one unit located in the northern region, on the banks of the Rio Negro in Manaus (located at

latitude $\lambda = 3°\ 08'47''$ and longitude $\varphi = 59°\ 57'\ 15''$) – the Issac Sabbá refinery (REMAN). The total emission of each pollutant of the refinery is given by equation (2):

$$E_{p(i,j)} = EP_p \times V, \tag{2}$$

where $E_{p(i,j)}$ represents the emission of the pollutant P, in grams per hour (g h$^{-1}$), $EP_p$ is the emission factor of pollutant P in grams per liter (g L$^{-1}$) and V is the volume of refined petroleum in liters per hour (L h$^{-1}$).

According to the National Petroleum Agency (ANP, 2014b), REMAN has as average production of approximately $1.7 \times 10^6$ liters of petroleum per hour. The emission factors of the pollutants have been admitted from Presidente Bernardes Refinary (RPBC), located in Cubatão, São Paulo, Brazil. The VOC speciation profiles used have also been obtained from the EPA (1990).

The total emissions for CO, $NO_x$, $SO_2$, PM and VOCs for the stationary and mobile sources that represent the

C0 scenario are shown in Table S1 in the electronic supplementary material. For example, Figure 3 shows the contribution, in percentage, of $NO_x$ and $SO_2$ emissions from all anthropogenic sectors considered





**2.4 Numerical Scenarios**

The investigations were defined by five numerical experiments (Table 4) to the study the influence of mobile and stationary emissions in the region, which are:

▪ Scenario C0, considering the main sources in the region (biogenic natural emission, vehicle, TPPs and REMAN refinery) representing the emission inventory of current conditions of the region;

▪ Scenario C1, numerical experiment with only biogenic natural emission, which simulates an atmospheric condition of an environment without the interference of human activities;

▪ Scenario C2, simulation with natural and vehicular emissions, intended to evaluate how the anthropogenic

emissions characterized on solely vehicular sources affect the atmospheric chemistry;

▪ Scenario C3, considering natural and TPPs and refinery anthropogenic emissions, aimed at assessing the impact of stationary sources on air pollutant emissions;

▪ Scenario C4, simulation including natural and the double of mobile and stationary emission sources, and double of the urban area of Manaus, which aimed at simulating an environment presenting a growth of the

regional population and urban area, as well as increased anthropogenic emissions. The urban expansion has been concentrated in the agricultural and forest areas, which was preserved the water resources, as well as forest reserves and parks around the city of Manaus.

The meteorological fields and air pollutants obtained from the simulation of scenario C0 have been compared to the ground-based data observed (see Figure 1). The analysis of meteorological variables has been conducted

based on the Pielke's skill ($S_{pielke}$) (Pielke, 2002; Pielke, Hallak and Perreira Filho, 2011), Pearson's correlation coefficient (r), and mean bias (MB). For the air pollutants, in addition to r and MB, two statistical indexes have been used, the mean normalized bias error (MNBE) and the mean normalized gross error (MNGE). Such parameters have often been used in several studies to assess the performance of atmospheric models (e.g., Tie et al., 2007; Han et al., 2009; Tuccella et al., 2012). Table 5 shows a summary of the statistical indices used to

evaluate the model performance.

**3 Results**

**3.1 Evaluation of the baseline scenario**

**3.1.1 Meteorology**

Figure 4 shows the comparison between the observed and the simulated values for temperature and relative

humidity at Colégio Militar, IFAM, T1 and T3 stations, corresponding to the periods of March 17 - 18, 2014. Overall, there is a good representation of the temporal evolution of the temperature both for the average value and the daily minimum. The T1 station has the highest peak of temperature: about 36 °C, which is the highest among the monitoring points compared to simulations. However, the model has proven to present difficulties in simulating the maximum temperature, mainly for T1 station, which is located in the central part of the urban area

of Manaus. For the IFAM station, which is also a central site, diurnal peak and minimum night temperatures were weakly represented. For all stations, the model presents problems to capture the peak temperature. The relative humidity profiles show a good level of agreement of the simulation to the average values of Colégio Militar and T1 stations. However, IFAM and T3 sites have the greatest discrepancies among the results.





Regarding the statistical indexes presented in Table 6, the correlation coefficient (r) has provided satisfactory

results for all stations, with the lowest values for temperature (0.87) and humidity (0.71) associated with T3 station

and the highest values 0.91 and 0.89 for Colégio Militar, respectively. According to Pielke's parameter skill, a

good model performance occurs for index values of less than 2. In this case, the performance of the model is

satisfactory for most of the stations, except for temperature at T1 site ($S_{pielke}$ = 2.9) and relative humidity at IFAM

($S_{pielke}$ = 3.7) and T3 ($S_{pielke}$ = 3.9) stations. According to the mean bias, it has been observed that the simulation,

in general, underestimates the majority of observed values, mainly for temperature (T1) and relative humidity

(IFAM and T3).

### 3.1.2 Air pollutants

In relation to air pollutants, T1 is the only station that has air quality data and only for $NO_x$, CO and $PM_{2.5}$

concentrations. Figure 5 and Table 7 show the comparisons between the simulation (scenario C0) and the

observation of such pollutants. In terms of the $PM_{2.5}$, it has been observed, in general, that the model tends to

overestimate the values observed (MNBE > 0), whereas with CO and $NO_x$, the tendency is the opposite. They

underestimated the values in most of the simulation period (MNBE < 0). The most significant differences have

been observed during the peak hours and mainly for CO. The weakest performance of the model happens mainly

during the nocturnal peak, situated generally between 18:00 and midnight (local time). A possible explanation for

the poor performance of the model during nocturnal peaks of CO is the fact that the fire outbreaks were not

considered, which could have influenced the simulated values. The MB indicates that the three analyzed pollutants

present significant differences, with the value of $PM_{2.5}$ MB of 1.30 µg m$^{-3}$ against a 0.66 µg m$^{-3}$ average observed,

$NO_x$ with values of -26.2 ppb against an 88.7 ppb average observed, and CO presenting -135 ppb of MB against

a 382.6 ppb average observed. Similar to the Pearson coefficient, the model shows good correlations for $PM_{2.5}$,

$NO_x$ and CO with 0.72, 0.53 and 0.53, respectively. In addition, in order to assess the sensitivity of the model to

respond to different emission scenarios, numerical experiments were carried considering variations in emission

of ± 15% and ± 30% in relation to scenario C0 (biogenic natural emission + vehicle + thermal power plants +

REMAN refinery). Overall, it was observed that the model responds linearly to variations performed. Temporal

evolution of $PM_{10}$ and $NO_x$ concentrations is shown in Figure S1.

There are a few studies measuring the impact of the urban plume of Manaus on the pollutant concentration

downwind of the city. For example, CO concentrations of 90-120 ppb were observed by Martin et al., 2016b from

crosswind transects inside the planetary boundary layer (PBL) of urban plume of Manaus downwind. Considering

the scenario C0, representing the emission inventory of current conditions for the region, the values found in this

study stayed in the 84-207 ppb range, with most differences being observed in the first levels of the model. It is

not possible to say that the model is over-predicting the actual value, since it is associated to different periods. In

addition, the simulated concentrations are much lower than the CO concentrations observed in previous

measurements for the dry season in Amazonas, under strong effects of biomass burning emission (e.g., Crutzen

et al., 1985). $O_3$ mixing ratios reported by Kuhn et al. (2010) varied from 21 ppb in the adjacent background to

peak value of 63 ppb for $O_3$ within the PBL at 100 km distance from Manaus. In the simulated scenario C0, the

peak values were in the range 26-29 ppb within the PBL and 40 ppb above it. $O_3$ mixing ratios under influence of

anthropogenic pollution were also reported by Trebs et al. (2012) and were on average 31±14 ppb, with peak

values of 60 ppb at a distance of 19 km downwind of Manaus. Such results are in good agreement with the results





in this work, 10 – 43 ppb for the adjacent levels to 400 m at 10 km west of Manaus, considering the current conditions given by scenario C0.

In terms of previous numerical studies in the Manaus urban-influenced area, Kuhn et al. (2010) applied a Single Column chemistry and meteorological Model – SCM (Ganzeveld et al., 2002). Two numerical experiment were performed to assess the westward-moving plume, the first one performed by using anthropogenic emission fluxes from the EDGAR 3.2 emission database, but with the pollutant flux based on the $1° \times 1°$ grid point of the location of the city of São Paulo, increased by a factor of 7, due to the absence of a local emission inventory for

Manaus. In this scenario, the simulated CO, $NO_x$ and $O_3$ mixing ratios at approximately 400 m altitude and 10 km downwind were of approximately 140 ppb, 4 ppb, and 50 ppb, respectively. The impact of stationary sources was evaluated in a second scenario by the inclusion of four thermal power plants, adding a total of 1.5 kg $NO_x$ day$^{-1}$. As a result, the simulated $NO_x$ and $O_3$ mixing ratios changed to 30 ppb and 35 ppb, respectively. The effects of thermal power plants are also evidenced in this work by the low concentration of $NO_x$ in the scenarios C1 and C2

(which do not include the stationary emission sources), compared to C0 (current conditions).

### 3.2 Scenario emissions

The discussions on the simulation results related to scenario emissions were analyzed in the lowest model level and divided in three topics, that is, the analysis of the mean and peak behavior of pollutants, spatial analysis, and temporal evaluation described below.

**3.2.1 Mean and peak behavior of pollutants**

Identifying the differences between scenarios in atmosphere models with elevated complexity is not a straightforward task, due to the large number of components involved, with each of them partially contributing to the evolution of the concentrations in both time and space. In this sense, its impact should be evaluated in terms of average value as peak. For this purpose, the methodology for microphysical variable proposed by Martins et

al. (2009) has been adapted to evaluate the air pollutants in this study. There are two average properties. The first is the Spatial Average-Temporal Average (SATA), which represents a mean value average both spatially and over time; and the second one is the Spatial Peak-Temporal Average (SPTA), which corresponds to the spatial peak average over time. The calculation of the values was based on 87% of the study grid, removing 5 points of the grid in each extremity. The values recommended by Skamarock et al. (2008) were used in order to reduce the

effect of lateral boundary conditions. It is important to emphasize that due to the large number of chemicals involved in the VOCs, the evaluation has been conducted by summing all output model compounds.

The impact of the different scenarios evaluated according to the average (SATA) and peak (SPTA) concentrations are organized in Table 8. As expected, the lower concentrations of SATA and SPTA of all pollutants are those obtained from natural conditions (scenario C1). Similarly, the futuristic scenario (C4) has

shown the highest concentrations simulated for the parameters applied. Considering the current conditions of the region (C0), scenario C4 represents an increase of approximately 35%, 3%, 62%, 4%, 16%, 42% and 41% (SATA), and 45%, 63%, 88%, 45%, 109%, 60% and 56% (SPTA) for the pollutants $NO_x$, CO, $SO_2$, $O_3$, VOCs, $PM_{2.5}$ e $PM_{10}$, respectively. In terms of the individual contribution by sources, it was observed that the emission contributions from stationary sources, predominantly TPPs (scenario C3), were greater than the mobile sources

(C2) for all analyzed pollutants, except for CO. For instance, the average concentration is more than double for



NOx and about one order of magnitude for peak values of PM10. This indicates that the TPPs are mainly responsible for the high concentrations of most chemical species in the grid. The influence of matrix change is discussed in a companion work of Medeiros et al., 2016 and more details can found in this paper.

### 3.2.2 Spatial Analysis

In order to analyze the plume of Manaus city, the spatial distributions of pollutants evaluated have been performed. The scenario distributions have indicated that the pollution plume from Manaus could have great impact on the surrounding area, with a predominance to the west and southwest directions. The impact is both on the average values and the peak values. It has been observed that the high values of emission rates from TPPs significantly contribute to the increase in the air pollution plume area of Manaus during its spread. The influence is predominant in adjacent regions, but it can be extended over more than a hundred kilometers to the west and southwest of the city, to areas that were dominated by the original forest. As an example, Figure 6 illustrates the spatial distributions of PM2.5 concentration for 22 hours local time, on March 18, 2014.

For PM2.5, considering the contour lines greater than or equal to 5 µg m$^{-3}$, it can be observed that the largest fractions of plume covering the area as shown in Figure 6 were 3,024 km$^2$ (C0), 1,386 km$^2$ (C3), and 6,102 km$^2$ (C4). Based on the plume covering the area and on the contour lines established, for the futuristic scenario, an area 16-fold greater than the Manaus urban area (MUA) has been observed, which is approximately 377 km$^2$. In addition, it has also been noted that there has not been any impact on the experiment by vehicular sources (C2) within the defined contour in this timetable. The values of the influence in an area and their contour lines for other pollutants are summarized in Table 9 as a function of MUA.

### 3.2.3 Temporal evaluation

Considering the importance of assessing the temporal development of the concentrations for a locality, two distinct points have been selected. The first one located within the city of Manaus near the T1 station, and the second in a predominant direction of the plume propagation at approximately 84 km southwest (T3 station), which can be identified in Figure 1. It is important to emphasize that the points were chosen in accordance with the highest concentrations of special distribution observed. From this evaluation, the important role of the stationary sources in the results of the concentrations of air pollutants has become evident. For example, based on Figure 7, the highest NOx concentrations are found in scenario C4, followed by C0 and C3, following the trends presented by SATA and SPTA. Even in locations that are distant from the city of Manaus, high concentrations when compared to concentrations in conditions not influenced by human activities (scenario C1) have been covered. In addition, it has been observed that all scenarios with the participation of TPPs have a significant impact on the direction of the plume spread, while Scenarios C1 and C2 remained with their concentrations near zero. This observation reinforces the fact that the purely vehicular source plume could not produce significant impact over long distances.

### Conclusions and Discussions

The priority in this study was to represent the daily cycle of emissions, which promote the development of an urban plume downstream of Manaus urban area. In this sense, it is important to investigate the relative contribution of the mobile and stationary sources and also the potential impact of the energy matrix change in the area impacted





by urban pollution plume. However, the conclusions presented here should see with caution in order that it involves only two specific days and they are not representative for all year long.

According to the statistical criteria used for the type of model considered, comparing the observed data with the simulation results to the current state of emissions defined by the baseline scenario (C0), we have concluded that the WRF-Chem performance can be considered satisfactory due to the representative criteria for most meteorological fields and air pollutants.

Based on the analyses of observed average and peak concentrations of pollutants, as well as the spatial and
temporal distribution of numerical studies, it is clear that stationary sources have an important role in the contribution of human activity in Manaus. The exception is given for carbon monoxide, which has shown little significant contributions. For a comparative evaluation of the results presented herein and those in literature, it has been observed that there is no parallelism with studies of other parts of the world that have an urban area in a vast tropical forest cover. Several studies have been identified in which the contribution of thermal power plants
is higher than the vehicle sources without, however, presenting preserved natural environment conditions, as occurs in Manaus. Studies such as Reddy and Venkataramn (2002) from India show that the main emission of a particulate matter and sulfur dioxide originates from fossil fuel power plants. In this example, $SO_2$ emissions by power plants are responsible for over 60% of the country's emissions. Another example, in this case in China, Zhang et al. (2012) conclude that most $PM_{2.5}$ (nitrate and sulfate) sources are from the energy sector, mainly power
plants, exceeding the combined contribution of industrial, residential and transport sources. Also from China, results by Huang et al. (2011) estimate that stationary sources contribute in approximately 97%, 86%, 89%, 91% and 69% of the total emissions of $SO_2$, $NO_x$, $PM_{10}$, $PM_{2.5}$ and VOCs, respectively.

In the specific case of Manaus, it is important to emphasize that the only hydroelectric plant located in the region that is able to bring electricity to Manaus is the Balbina hydroelectric power station. It has an average
capacity of only 250 MW and it contributes to less than 15% of the electricity used in Manaus (ANEEL, 2014). Therefore, the largest source of electricity for the region comes from power plants burning petroleum fuels, particularly fuel oil and diesel, with higher emission factors if compared, for example, to natural gas.

Another relevant issue is the high electricity consumption of Manaus. According to data from the Brazilian Energy Research Company (EPE) (www.epe.gov.br/), the electricity consumption per capita in Brazil is 2.5 MWh
per year, whereas in the city of Manaus, the consumption is 7.2 MWh, about 3-folds higher. Two aspects make the region highly dependent on electricity. The first is the incentive policies for the consolidation of the industrial area in Manaus in order to foster the economic development of the region. This resulted in a region with a variety of industrial centers, with a high demand for electricity. The second aspect is the fact that the city is located in a humid equatorial climate region, dominated by high heat and humidity, as well as little ventilation and torrential
rains throughout the year. Such environmental characteristics induce high electricity consumption by both residential and commercial sectors, resulting in the greater burning of fossil fuel power plants that intensify the concentrations of air pollutants.

The spatial distributions of these scenarios indicated that the pollution plume from Manaus could have a great impact on the surrounding areas, mainly in west and southwest directions, reaching hundreds of kilometers from
the city. Since most thermal power plants and the REMAN refinery are located near the banks of the Negro and Solimões rivers, the plume transportation could be influenced by the circulation of river breezes, which defines




the trajectory of pollutants. Although the breeze effect is not focused on this paper, this evaluation has been confirmed by other studies for the large rivers in the Amazon, for instance, Santos et al. (2014).

Finally, in order to evaluate the potential urban growth in that region, a futuristic scenario has been designed.
In summary, this scenario has shown the impact of a possible increase of mobile and stationary emissions in the study region, including the expansion of the urban area. The approach of futuristic scenarios has been studied by several methodologies, mainly in Asia (Zhou et al., 2003; Ohara et al., 2007). In all cases, the increase in air pollution concentrations could be observed if the current conditions of the energy matrix were maintained. However, there is the possibility of reductions and a decrease of emission rates by using sustainable energies.

**Acknowledgements**

 The authors would like to gratefully acknowledge the measurements of the Project Observation and Modelling of the Green Ocean Amazon (GoAmazon) and the REMCLAM network for climate Change Amazon from the University of the State of Amazonas (UEA) that contributed to this study. This work was supported by CNPq (Conselho Nacional de Desenvolvimento Científico e Tecnológico, process 404104/2013-4), CAPES
(Coordenação de Aperfeiçoamento de Pessoal de Nível Superior) and Araucária Foundation.

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



**Table 1. Physical parametrization used for this study.**

| Process | Reference |
| --- | --- |
| Microphysics | Milbrandt-Yau (2005) |
| Surface Layer | MM5 (Zhang and Anthes, 1982) |
| Soil-Land Parameterization | Noah LSM (Chen and Dudhia, 2001) |
| Boundary Layer | YSU (Hong and Dudhia, 2003) |
| Shortwave Radiation | Dudhia (Dudhia, 1989) |
| Longwave Radiation | RRTM (Mlawer et al., 1997) |





**Table 2. Emission factors for CO, NOx, PM, SO2, and VOCs, in grams per kilometer, for different vehicle type/fuel combinations.**

| Vehicle type | Fuel burned | CO | NOx | PM | SO2 | VOCs (evaporative) | VOCs (liquid) | VOCs (exhaust) |
|---|---|---|---|---|---|---|---|---|
| Light-duty vehicles | Gasohol | 5.43 | 0.34 | 0.15 | 0.03 | 0.17 | 2.00 | 1.24 |
| | Ethanol | 12.00 | 1.12 | 0.15 | 0.01 | 0.04 | 1.50 | 2.12 |
| | Flex | 5.13 | 0.32 | 0.15 | 0.02 | - | - | 0.15 |
| Heavy-duty vehicles | Diesel | 4.95 | 9.81 | 0.44 | 0.61 | - | 0 | 2.48 |
| Motorcycles | Gasohol | 9.15 | 0.13 | 0.05 | 0.01 | - | 1.40 | 2.37 |



**Table 3. Emission factor of TPPs per fuel type.**

| Fuel type | Natural gas | | Fuel oil | | Diesel | |
| --- | --- | --- | --- | --- | --- | --- |
| | $(g\ 10^6\ L^{-1})$ | | $(g\ L^{-1})$ | | $(g\ L^{-1})$ | |
| Pollutant | EPA | CETESB | EPA | CETESB | EPA | CETESB |
| CO | 384 – 1568 | – | 0.6 | – | 4.9 – 2.4 | – |
| PM | 121.6 | 48 – 219 | 0.05 – 1.2 | 0.84 – 1.45 | 0.59 – 1.6 | 0.24 |
| $NO_x$ | 512 – 4480 | 2240 – 8800 | 1.2 – 6.6 | 6.6 – 8 | 18.3 – 54.1 | 2.4 |
| $SO_2$ | 9.6 | 9.6 | 17 – 18.8 | 18.2 – 19.2 | 5.8 – 18.2 | 17.2 |
| TOCs | 176 | 28 – 92 | 0.03 – 0.3 | 0.03 – 0.13 | 0.04 – 1.6 | 0.03 |

**Source: EPA, AP-42, 1998, 2010; CETESB, 2009.**





**Table 4. Summary of simulation scenarios.**

| Scenarios | Emissions |
|---|---|
| C0 | Natural, vehicular and stationary (TPP and REMAN refinery) sources |
| C1 | Only natural sources |
| C2 | Natural and vehicular |
| C3 | Natural and stationary (TPPs and REMAN refinery) sources |
| C4 | Natural and the double of mobile and stationary sources |





**Table 5. Statistical indexes used to evaluate the model performance.**

| Indexes | Equation[a] |
|---|---|
| Pearson's Correlation Coefficient | $r = \dfrac{\sum_{k=1}^{N}(p_k - \bar{p})(o_k - \bar{o})}{[\sum_{k=1}^{N}(p_k - \bar{p})^2]^{1/2}[\sum_{k=1}^{N}(o_k - \bar{o})^2]^{1/2}}$ |
| Mean Bias | $MB = \dfrac{1}{N}\sum_{k=1}^{N}(p_k - o_k) = \bar{p} - \bar{o}$ |
| Root Mean Square Error | $RMSE = [\dfrac{1}{N}\sum_{k=1}^{N}(p_k - o_k)^2]^{1/2}$ |
| Skill of Pielke | $S_{Pielke} = \left|1 - \dfrac{\sigma_p}{\sigma_o}\right| + \dfrac{RMSE}{\sigma_o} + \dfrac{RMSE - BIAS}{\sigma_p}$ |
| Mean Normalized Bias Error | $MNBE = \dfrac{1}{N}\sum_{k=1}^{N}\left[\dfrac{(p_k - o_k)}{o_k}\right] \times 100$ |
| Mean Normalized Gross Error | $MNG = \dfrac{1}{N}\sum_{k=1}^{N}\left[\dfrac{|p_k - o_k|}{o_k}\right] \times 100$ |

a.      $p_k$ and $o_k$ are the predicted and observed value at time k, respectively.





**Table 6. Evaluation indexes for model performance of meteorological variables during the simulation period (baseline scenario C0).**

| Variable | Station | Mean$_{obs}$ | Mean$_{sim}$ | $\sigma_{obs}$ | $\sigma_{sim}$ | r | MB | RMSE | S$_{pielke}$ |
|---|---|---|---|---|---|---|---|---|---|
| Temperature (°C) | Colégio Militar | 29.6 | 28.8 | 2.3 | 1.7 | 0.91 | -0.8 | 1.3 | 1.9 |
| | IFAM | 28.2 | 28.5 | 2.8 | 1.8 | 0.88 | 0.3 | 1.5 | 1.6 |
| | T1 | 30.0 | 28.5 | 2.5 | 1.8 | 0.88 | -1.5 | 1.9 | 2.9 |
| | T3 | 26.6 | 26.5 | 2.6 | 2.5 | 0.87 | -0.1 | 1.3 | 1.1 |
| Relative humidity (%) | Colégio Militar | 73.6 | 70.7 | 9.1 | 8.1 | 0.89 | -2.9 | 5.1 | 1.7 |
| | IFAM | 80.4 | 72.1 | 10.5 | 7.2 | 0.89 | -8.3 | 9.7 | 3.7 |
| | T1 | 73.9 | 71.8 | 9.3 | 8.2 | 0.88 | -2.1 | 4.8 | 1.5 |
| | T3 | 89.8 | 79.7 | 10.6 | 8.7 | 0.71 | -10.1 | 12.6 | 3.9 |



**Table 7. Evaluation indexes for model performance of air pollutants during the simulation period (baseline scenario C0).**

| Variable | Mean$_{obs}$ | Mean$_{sim}$ | $\sigma_{obs}$ | $\sigma_{sim}$ | r | MB | MNBE (%) | MNG (%) |
|---|---|---|---|---|---|---|---|---|
| PM$_{2.5}$ (ug m$^{-3)}$) | 1.30 | 1.96 | 0.81 | 0.93 | 0.72 | 0.66 | 50.8 | 57.3 |
| NO$_x$ (ppb) | 88.7 | 62.5 | 53 | 52.7 | 0.53 | - 26.2 | -29.5 | 47.4 |
| CO (ppb) | 382.6 | 247.3 | 296.3 | 93.3 | 0.53 | - 135.3 | -35.4 | 53.1 |





**Table 8. Spatial Average-Temporal Average (SATA) and Spatial Peak-Temporal Average (SPTA) of the concentration of trace gases and particles, in the various scenarios[a].**

|  | Variable | Unit | C0 | C1 | C2 | C3 | C4 |
|---|---|---|---|---|---|---|---|
| $NO_x$ | SPTA | ppb | 208.68 | 1.27 | 40.81 | 198.21 | 302.85 |
| $NO_x$ | SATA | ppb | 1.02 | 0.11 | 0.42 | 0.80 | 1.38 |
| CO | SPTA | ppb | 291.51 | 83.37 | 261.74 | 174.68 | 476.13 |
| CO | SATA | ppb | 84.75 | 81.20 | 83.77 | 82.31 | 87.60 |
| $SO_2$ | SPTA | ppb | 278.96 | 0.10 | 1.83 | 275.46 | 525.78 |
| $SO_2$ | SATA | ppb | 0.85 | 0.06 | 0.08 | 0.79 | 1.38 |
| $O_3$ | SPTA | ppb | 96.32 | 32.49 | 50.82 | 91.24 | 139.62 |
| $O_3$ | SATA | ppb | 26.49 | 23.70 | 25.01 | 25.68 | 27.60 |
| VOCs | SPTA | ppb | 1814 | 35.66 | 77.87 | 1853 | 3799 |
| VOCs | SATA | ppb | 19.31 | 15.67 | 16.31 | 19.17 | 22.45 |
| $PM_{2.5}$ | SPTA | $\mu g\ m^{-3}$ | 32.51 | 0.35 | 3.49 | 32.38 | 51.89 |
| $PM_{2.5}$ | SATA | $\mu g\ m^{-3}$ | 0.48 | 0.19 | 0.25 | 0.42 | 0.68 |
| $PM_{10}$ | SPTA | $\mu g\ m^{-3}$ | 40.72 | 0.35 | 4.51 | 40.47 | 63.36 |
| $PM_{10}$ | SATA | $\mu g\ m^{-3}$ | 0.51 | 0.19 | 0.26 | 0.44 | 0.72 |

[a] baseline scenario (emissions by biogenic, mobile and stationary sources); C1, only natural emissions; C2, scenario referring to natural and mobile emissions; C3, scenario representing natural and stationary emissions; C4, futuristic scenario that includes natural emissions and double emissions from mobile and stationary sources.




**Table 9**. Area influenced by the pollution plume, represented as a function of MUA, for all simulated scenarios.

| Variable | C0 | C1 | C2 | C3 | C4 |
|---|---|---|---|---|---|
| $NO_x$ (contour lines $\geq$ 30 ppb) <br> 8 LT – 18/03/2014 | 6 | - | 2 | 4 | 8 |
| CO (contour lines $\geq$ 200 ppb) <br> 8 LT – 18/03/2014 | 3 | - | 3 | 0.1 | 6 |
| $SO_2$ (contour lines $\geq$ 50 ppb) <br> 20 LT – 18/03/2014 | 1 | - | - | 1 | 2 |
| $O_3$ (contour lines $\geq$ 50 ppb) <br> 17 LT – 18/03/2014 | 20 | - | 3 | 15 | 26 |
| VOCs (contour lines $\geq$ 100 ppb) <br> 8 LT – 18/03/2014 | 6 | - | - | 6 | 11 |
| $PM_{2.5}$ (contour lines $\geq$ 5 $\mu g\ m^{-3}$) <br> 22 LT – 18/03/2014 | 8 | - | - | 4 | 16 |
| $PM_{10}$ (contour lines $\geq$ 5 $\mu g\ m^{-3}$) <br> 22 LT – 18/03/2014 | 8 | - | 0.1 | 4 | 17 |



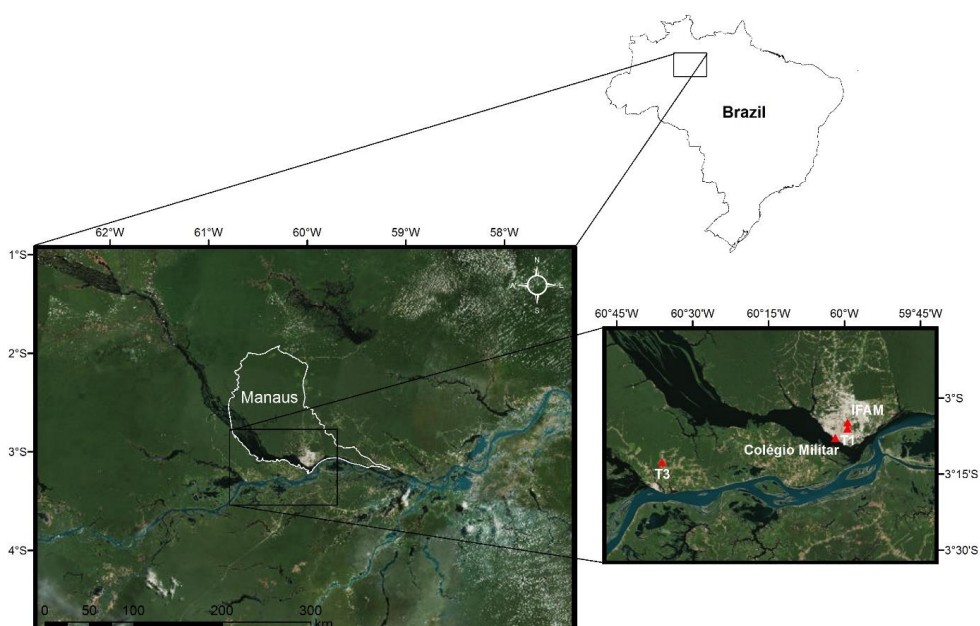

**Figure 1.** Geographic location of the study area, meteorology stations (Colégio Militar, IFAM and T3) and air quality station (T1). The white contour line shows the delimitation of the city of Manaus.





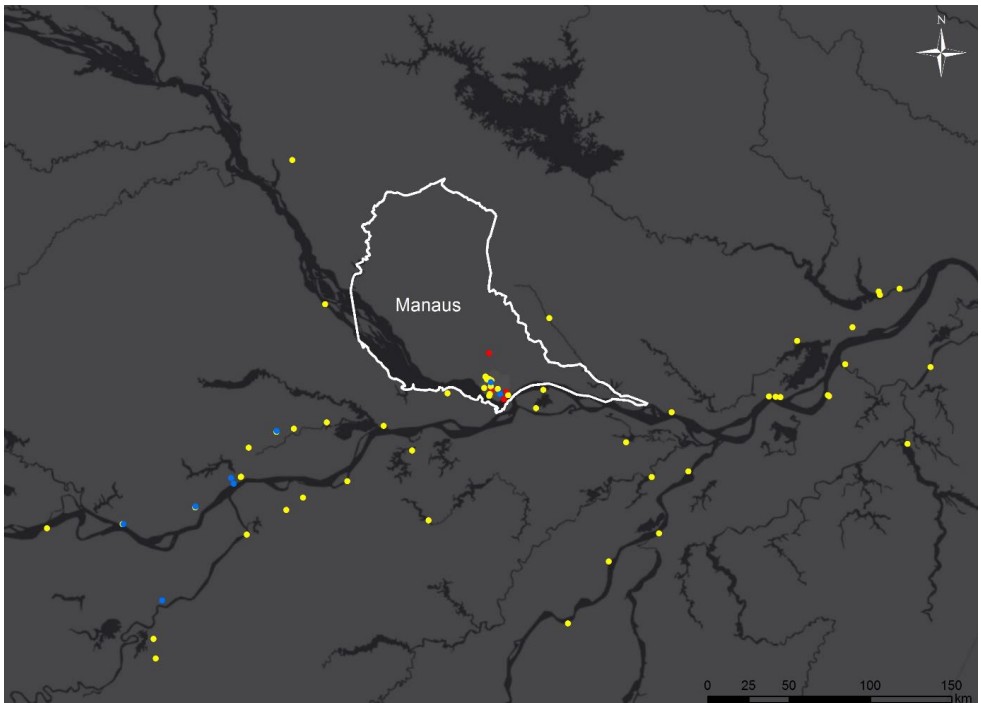

**Figure 2. Spatial distribution of diesel (yellow), fuel oil (red) and natural gas (blue) TPPs on the study grid, and the border of Manaus (white).**



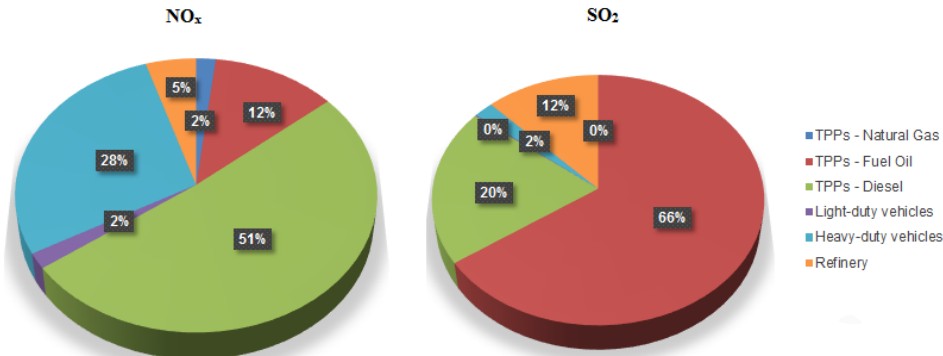

**Figure 3. NO$_x$ and SO$_2$ emission contribution from all anthropogenic sectors considered.**





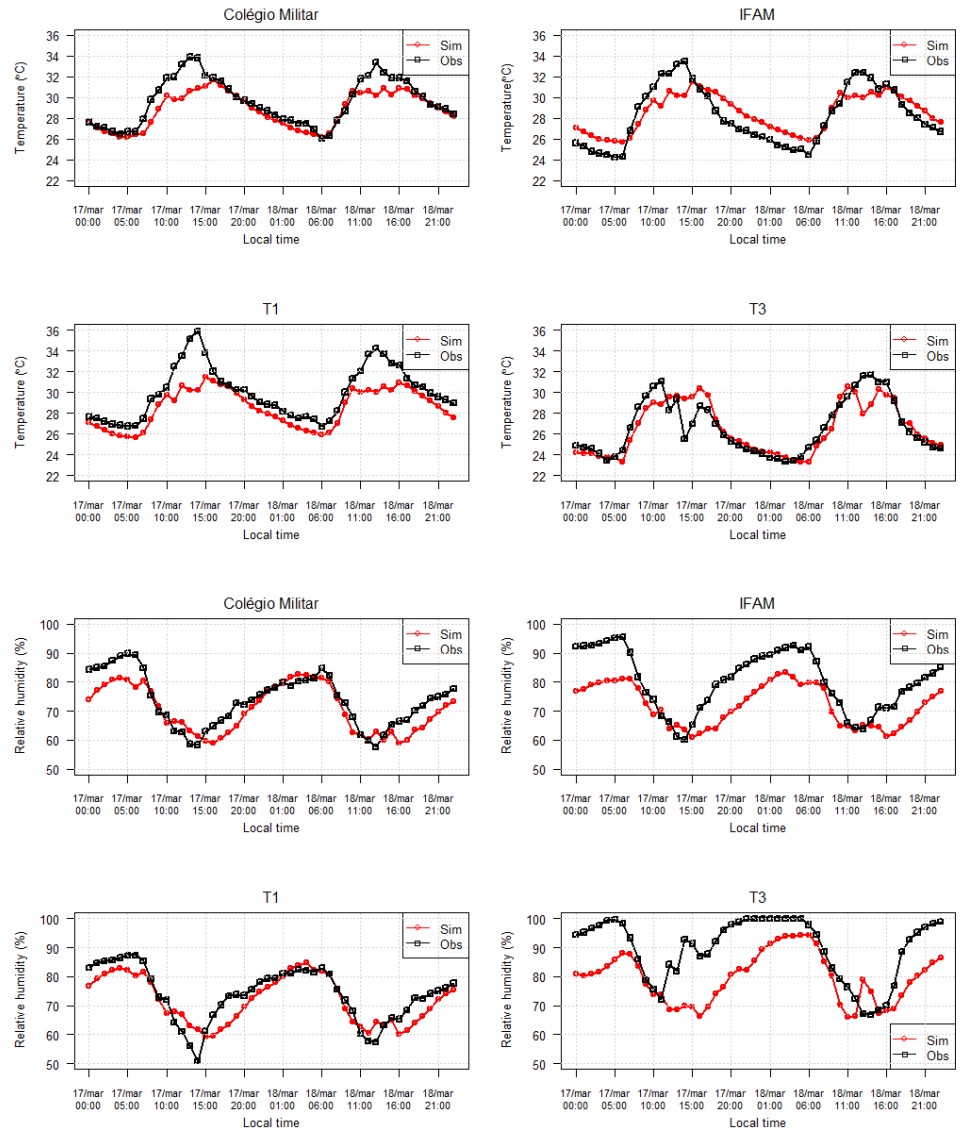

**Figure 4.** Temporal evaluation of temperature and relative humidity simulated and observed for Colégio Militar, IFAM, T1 and T3 stations.




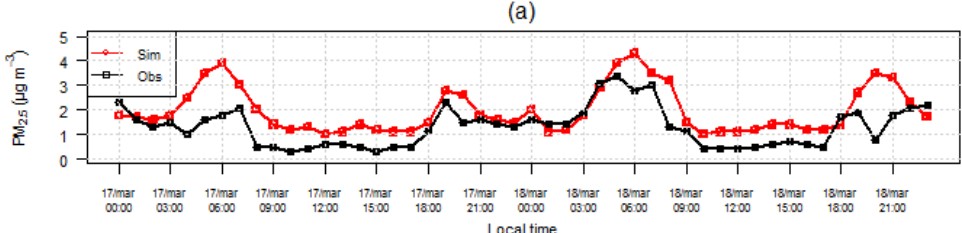

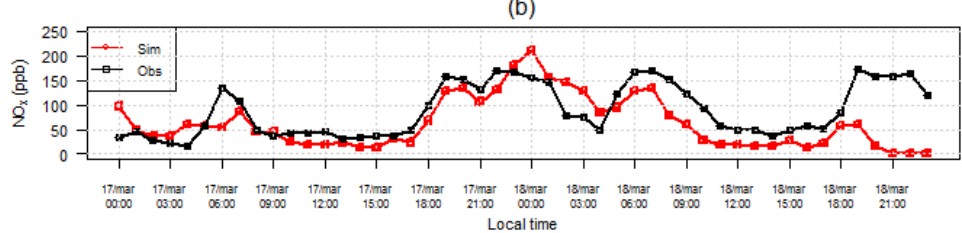

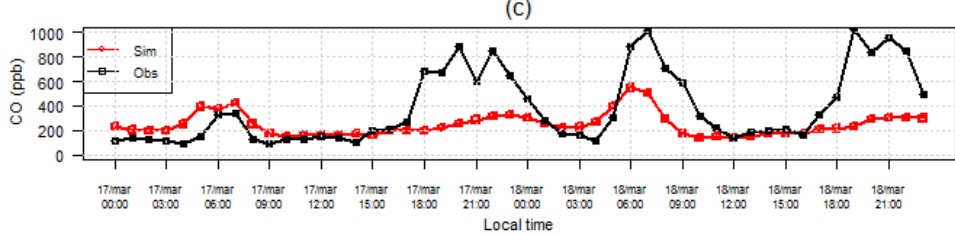

**Figure 5. Temporal evaluation of simulated and observed values for the concentration of pollutants in T1 station: (a) PM2,5, (b) NOx and (c) CO.**





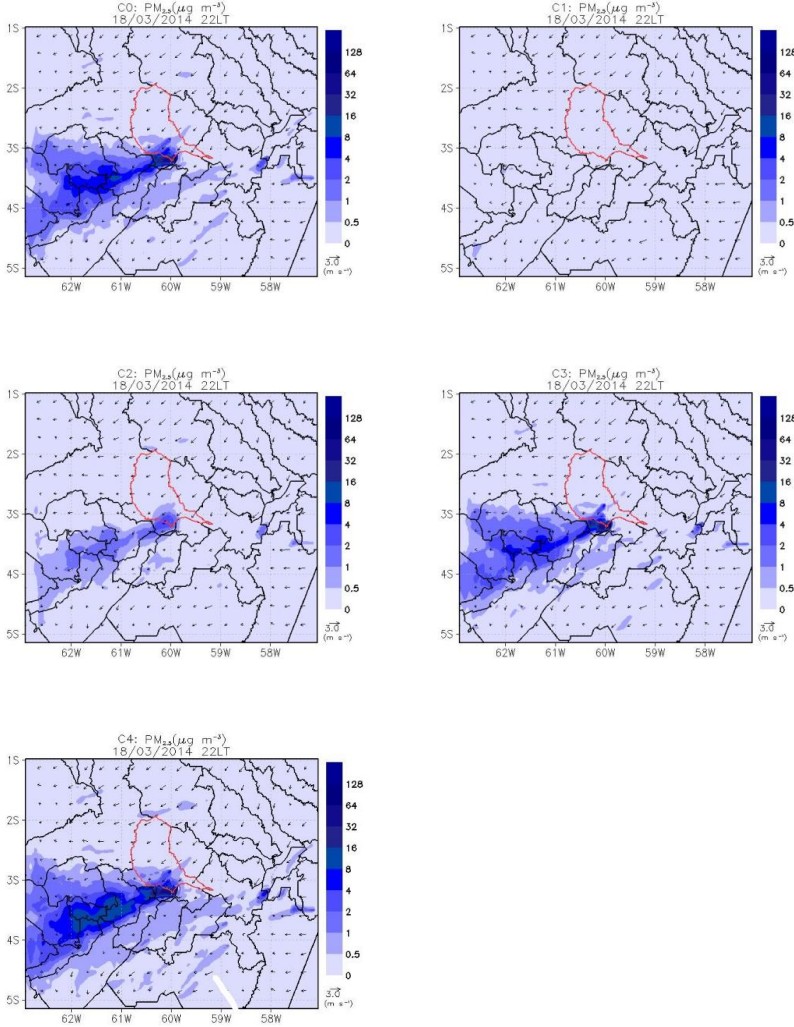

**Figure 6. Spatial distribution of the scenarios studied for PM$_{2.5}$ concentration, calculated at 22 hours, local time, on March 18, 2014. The black contours illustrate the delimitations of the municipalities on the grid, and the red one represents Manaus.**





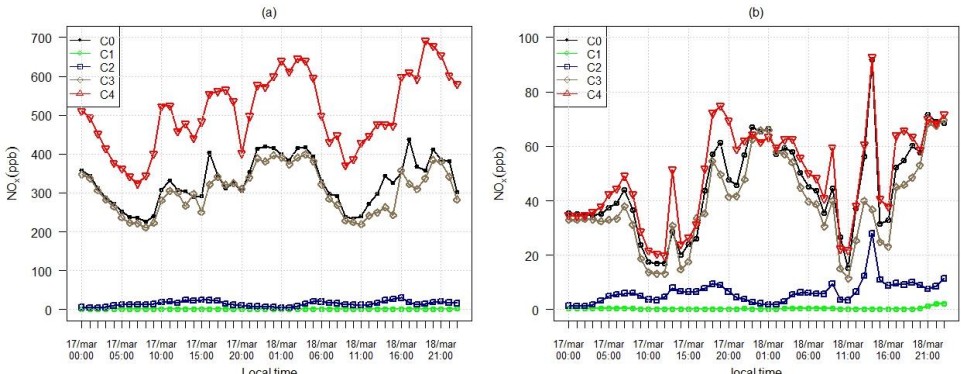

**Figure 7.** Temporal evaluation of NO$_x$. (a) represents the concentration within Manaus, while (b) refers to the point towards the pollution plume about 84 km southwest.