# Peer review of "Contributions of mobile, stationary and biogenic sources to air pollution in the Amazon rainforest: a numerical study with WRF-Chem model"

_Atmospheric Chemistry and Physics, 2016_

## Referee Comment (RC1) · Anonymous Referee #1 · 16 Feb 2017

**General Comments**

Anthropogenic emissions of gases and particulate matter are thought to adversely affect pollutant levels within the city of Manaus and the surrounding pristine rainforest. However, Manaus emissions inventories have been a major source of uncertainty in modeling studies examining processes affecting atmospheric composition in the Amazon Basin. Rafee et al. aim to improve understanding in three areas: (1) the relative contribution of stationary and vehicular sources to pollution levels in Manaus and the surrounding forest; (2) the location and area of forest affected by pollution in the Manaus plume; and (3) the potential impact of future increases in Manaus emissions on pollution levels in the city and surrounding forest. They accomplish this by conducting high-resolution WRF-Chem model simulations of a region surrounding Manaus for scenarios with current emissions (base case); only biogenic, stationary, or vehicular sources; and a future scenario doubling emissions and urban land use area. They evaluate the simulations against ground-based meteorological and air quality observations, determine the typical direction and extent of the Manaus plume, and compare simulated peak and average pollutant levels for the different scenarios. They conclude that stationary sources are the largest contributors to observed gas and particulate amounts, except for CO, for which vehicles are the most important source. The Manaus plume can extend hundreds of kilometers in the west and southwest direction. The future scenario increases average pollutant concentrations by up to 62% and peak values up to 109% relative to present emissions and land use.

The study is of great interest to those wishing to conduct modeling studies of Manaus and the Amazon Basin, as it provides an improved anthropogenic emissions inventory for Manaus and evaluates it against chemical observations. However, the authors themselves state the caveat that the present study only simulated two case study days during the rainy season due to lack of observations. It provides a methodological advance in emissions estimation for Manaus and new knowledge about the dominance of stationary sources in Manaus. It provides a foundation for future studies utilizing year-round ground-based chemical observations.

Nevertheless, the article could have broader interest and impact by expanding the discussion to the relative contributions of anthropogenic and biogenic sources and their interactions to levels of different pollutants. These anthropogenic-biogenic interactions are described as a motivation for the present study. Simulations were conducted with only biogenic emissions, providing information on the relative contributions of biogenic and anthropogenic sources that needs to be elaborated in the text. See my specific comments below.

Finally, I recommend to the editor that before publication the manuscript be thoroughly proofread for correct English usage and grammar. The manuscript is generally understandable, but there are numerous grammatical errors, only a few of which I have listed in the technical comments below.

**Specific Comments**

Title - For better alignment with the objectives of the study, I recommend altering the title to "Contributions of mobile, stationary, and biogenic sources to air pollution in the Amazon rainforest: a numerical study with the WRF-Chem model."

Abstract - I recommend including results on the relative impacts of anthropogenic and biogenic emissions on pollutants.

14 Please clarify the purpose of the future scenarios, e.g. "as well as a future scenario to assess the potential air quality impact of doubled anthropogenic emissions."

28 300 Million tons of coal equivalent relative to what amount previously?

32-34 Understanding the predominant sources of each pollutant is key to designing successful regulatory policy. Since this is a key objective in the study, mention this in the first paragraph.

52 Impact of anthropogenic emissions of air pollutants on what?

58 I recommend stating again here why we want to know the "participation of each type of emission source on air pollution" (e.g. regulatory purposes) to underline the present study's importance.

84 State here what the four measurement sites observed (e.g. meteorological parameters, air pollutant levels)

104 Why were these WRF-Chem physics options chosen? Some justification should be included here. Were simulations with different options conducted and evaluated against observations to find the best representation? Were they based on previous published studies?

134 It seems to me that initializing the model chemical fields with observed chemical profiles where available would provide a better result than using mid-latitude Northern Hemisphere profiles. How do observed profiles (e.g. from aircraft campaigns) compare with the WRF-Chem default profiles?

151 Please define "gasohol" in the text

153 How reasonable an assumption are emission factors measured in São Paulo? Do they depend on meteorological conditions?

177 Please include some examples of "transport and communication areas"

192-196 Please explain why you used EPA rather than CETESB TPP emission factors.

225 How do the emissions estimated in the present study compare with previous studies (previously existing inventories for Manaus, global inventories)?

255-269 Include numerical values indicating "good representation of the temporal evolution..", "diurnal peak and minimum night temperatures were weakly represented.", "relative humidity profiles show a good level of agreement..", and "IFAM and T3 sites have the greatest discrepancies among the results.", and "According to the mean bias.."

276-277 Provide values for MNBE

280 What "fire outbreaks"? Do you mean that there was fire influence during this period and fire emissions were not included in the simulations? On lines 93-95 you state "Another important factor when choosing the period of study was the low incidence of biomass burning wildfires in the rainy season."

288-289 Would you expect a similar linear result if you varied mobile and stationary sources independently?

290-294 Is the range of 84-207 ppb calculated at the same altitude the aircraft sampled at, or in the first model level, where higher CO values would be expected?

319 Do peak values have disproportionately higher impacts on public health? That is, are there threshold values of $PM_{2.5}$, $NO_x$, $O_3$ for health and ecosystem adverse impacts?

323-324 Unclear. Do you mean "In this sense, their impacts on peak average values should be evaluated."

345 Why was the contour of $PM_{2.5} \geq 5$ µg m$^{-3}$ chosen? Is it associated with some threshold of human health impact?

376-377 I recommend mentioning again that mobile/stationary emissions partitioning is important for designing regulations

379 How would you expect the results to change in the presence of biomass burning emissions?

422 Mention numbers again, e.g. "In all cases, an increase in air pollution concentrations (X%-X%) could be observed.."

**Technical Comments**

11 Insert "on air pollution" after "impact of the emissions from mobile and stationary sources."

14 Remove "Results show that"

16 Remove "ones"; I recommend replacing all instances of "futuristic scenario" with "future scenario."

17-18 Replace two instances of "has shown" with "showed"

19 Insert "transported" after "predominantly."

31 Alter "oxide nitrogen" to "nitrogen oxides"

33-34 Alter "technologies, which guarantee" to "technologies that guarantee"

37 Alter "the complex combination of the latter" to "their complex combination"

38 Alter "the transportation and the combination of atmospheric pollutants" to "the transport and interactions of atmospheric pollutants"

41-42 Alter "aspects of the pollution impact. For instance, pollution episodes.." to "aspects of the pollution impact, such as pollution episodes.."

43 Alter "transportation" to "transport"

44 Alter "effects on" to "effects of"

46 Alter "Such studies are not capable of investigating the impact of isolated urban plumes." to "Such studies did not investigate the impact of isolated urban plumes."

47 Alter "in the last decades" to "in recent decades"

58 Alter "Bela et al., 2014" to "Bela et al., 2015."; Alter "Therefore, such studies" to "These studies".

62 Insert "pollutant concentrations above" before "the preserved forest region"

69 Alter "Participation of mobile and stationary sources in.." to "Participation of mobile and stationary anthropogenic sources and biogenic sources in.."

76 Alter "scope" to "region"

77 and 215 Remove "$\lambda =$" and "$\varphi =$"

81 Alter "represents" to "is part of"

101-102 Move "simultaneously" to before "predicts"

113 Delete "side"

118 Insert "reactions" after "photolysis"

121 Insert "masl" after "height"

150 Alter "grid study" to "study grid"

200 Alter "Northern region" to "North region"

202-203 Alter two instances of "admitted" to "assumed"

207 Alter "greater" to "greatest"

214 Alter "northern region" to "North region"

273 Remove "In relation to air pollutants"

276-277 Alter "with CO and $NO_x$, the tendency is the opposite. They underestimated.." to "with CO and $NO_x$, the simulations underestimated.."

288-289 Move "Temporal evolution of $PM_{10}$ and $NO_x$ concentrations is shown in Figure S1." before "Overall, it was observed that the model responds linearly to variations performed."

300-303 Insert "downwind of Manaus" after "$O_3$ mixing ratios."

313 Insert "significant" before "effects"

314 Alter "in this work" to "in the present study"

323 Alter "its impact" to "their impact"

328-329 Alter "in each extremity. The values recommended by Skamarock et al. (2008) were used in order to reduce the effect of lateral boundary conditions." to "in each extremity, which were the values recommended by Skamarock et al. (2008) to reduce the effect of lateral boundary conditions."

332 Insert "spatial" before "average"

335 Alter "parameters applied" to parameters considered"

345 Alter In order to analyze the plume of Manaus city, the spatial distributions of pollutants evaluated have been performed." to "In order to analyze the spatial extent and location of impact of the plume of Manaus city, the spatial distributions of pollutants have been evaluated."

347-348 Alter "The impact is both on the average values and the peak values." to "The impact is both on the spatial average and spatial peak values."

375 Insert "not present in global inventories" after "daily cycle of emissions

427 "Network for Climate"

Table 1 - Caption - Alter "Physical parameterization used for this study." to "WRF-Chem physics parameterizations used for this study."; Alter "Soil-Land Parameterization" to "Soil-Land"

Table 3 - Caption – Alter "per" to "by"

Table 4 - Alter four instances of "natural" to "biogenic"

Table 9 - Insert units in parentheses after "Area"

Figure 2 - Could you combine Fig. 2 with Fig. 1 to save space?

Figure 5 - Caption – format subscripts for $PM_{2.5}$ and $NO_x$

---

## Short Comment (SC1) · 15 Mar 2017

The manuscript presents a contribution to the understanding of the dispersion conditions of primary and secondary air pollutants in Manaus, AM, Brazil, on clear sky days, for a Brazilian metropolis surrounded by primary tropical forest.

The main shortcoming is the very short simulation period due to the reduced availability of observational data. Despite this deficiency and considering the scarcity of publications presenting a documented inventory, I can make the recommendation for publication after a minor but indispensable revision.

Three main points that should be considered in the minor revision to be made by the

authors are indicated below:

1. Show each focus of heat and smoke on the map of South America during the days of the experiment, obtained from satellite imagery. One might ask what can be stated about the pollutant plume emitted from each heat source, especially around Manaus, with trajectories within the PBL. If the heat sources were many, or a focus is highlighted from the others by the intensity, what would be the axis of the dispersing smoke plume in the vicinity of Manaus? In addition, what is the concentration of transient pollutants on the area of interest? It is not enough to indicate in the text the absence of heat / smoke focus in the area because a distant focus may be the source of a pollutant feather that propagates through long trajectories to the vicinity of Manaus. Could you verify this?

2. Show if possible the spatial distribution of pollutant emission rates on the surface of the domain, graphically presenting the result of the inventory prepared and used. In addition, show corresponding two-dimensional figures. For the emission of CO2 and other pollutants of daytime variation, present maps of the emission rate every 3 hours over the 24 hour cycle.

3. Characterize the synoptic and mesoscale conditions present during the days of the numerical investigation. Satellite images are available? Look for the channels whose composition matches the image of air masses moving in the domain, thus doing to highlight the aerosol plumes if possible. Can this be done?

4. Spatial fields can be presented to characterize the synoptic condition: current lines and advection of equivalent potential temperature.

---

## Author Comment (AC1) · 4 May 2017

Dear Referee #1.

We would like to thank you for your insightful comments that enabled us to improve the quality of our manuscript. Detailed responses to your concerns are outlined below.

**Specific Comments**

**Referee Comment: Title - For better alignment with the objectives of the study, I recommend altering the title to "Contributions of mobile, stationary, and biogenic sources to air pollution in the Amazon rainforest: a numerical study with the WRF-Chem model."**

Author's Response: The authors accepted the suggestion, thanks.

**Referee Comment: Abstract - I recommend including results on the relative impacts of anthropogenic and biogenic emissions on pollutants.**

Author's Response: Thanks for the suggestion. We included the following sentence in the abstract: "The anthropogenic sources considered resulted an increasing in the spatial peak-temporal average concentrations of pollutants in 3 to 2,780 times in relation to those with only biogenic sources." (Page 1, L18-19).

**Referee Comment: L14 Please clarify the purpose of the future scenarios, e.g. "as well as a future scenario to assess the potential air quality impact of doubled anthropogenic emissions."**

Author's Response: We included "to assess the potential air quality impact of doubled anthropogenic emissions" after "as well as a future scenario". In addition, we decided to remove "which is twice the current emissions from mobile and stationary sources" after "The future scenario" (Page 1, L15-16).

**Referee Comment: L28 300 Million tons of coal equivalent relative to what amount previously?**

Author's Response: Amount previously: 1349 MtCE in 2001. We clarify this information in the manuscript. The following sentence we changed for: "). For instance, energy consumption in China has increased more than 300 million tons of coal equivalent

(MtCE), compared to the previously amount of 1349 MtCE. Most of this energy is produced from burning fossil fuels, mainly coal (Crompton and Wu, 2005)" (Page 1, L31-33).

**Referee Comment: 32-34 Understanding the predominant sources of each pollutant is key to designing successful regulatory policy. Since this is a key objective in the study, mention this in the first paragraph.**

Author's Response: Thanks for the suggestion. We mentioned the following sentence in the first paragraph: "Understanding the predominant sources of each pollutant is a key to designing successful regulatory policies to improve air quality and bring benefits to public health" (Page 1, L36-37).

**Referee Comment: 52 Impact of anthropogenic emissions of air pollutants on what?**

Author's Response: We apologize for that, it was a mistake. We corrected the sentence: "Thus, the region is a valuable laboratory, for example for studying the impact of anthropogenic emissions of air pollutants on atmospheric chemical composition" (Page 2, L54-55).

**Referee Comment: 58 I recommend stating again here why we want to know the "participation of each type of emission source on air pollution" (e.g. regulatory purposes) to underline the present study's importance.**

Author's Response: Thanks for the suggestion. We included the following sentence after "participation of each type of emission source on air pollution": "which is important for the formulation of regulatory public policies for air quality management" (Page 2, L63).

**Referee Comment: 84 State here what the four measurement sites observed (e.g. meteorological parameters, air pollutant levels)**

Author's Response: Done. We included "Four measurement sites of meteorological variables and air pollutants concentrations were available in different…" (Page 3, L89-90).

**Referee Comment: 104 Why were these WRF-Chem physics options chosen? Some justification should be included here. Were simulations with different options**

conducted and evaluated against observations to find the best representation? Were they based on previous published studies?

Author's Response: We added this information in the manuscript. We included the following sentence: "The physical options (Table 3) used were defined according to the recent options inserted in the model, as well as those already tested in simulations by other works (e.g. Vara-Vela et al., 2015). Furthermore, some combinations of physical parametrizations were tested and the combination that best represented the meteorological parameters observed (temperature and relative humidity) was used." (Page 3-4, L111-115).

**Referee Comment: 134 It seems to me that initializing the model chemical fields with observed chemical profiles where available would provide a better result than using mid-latitude Northern Hemisphere profiles. How do observed profiles (e.g. from aircraft campaigns) compare with the WRF-Chem default profiles?**

Author's Response: We agree that the observed profiles from aircraft campaigns would provide a better result. However, the simulations were performed when these campaigns have not initialized.

**Referee Comment: 151 Please define "gasohol" in the text**

Author's Response: Done. We included the following sentence in the manuscript: "..gasohol (a mixture of gasoline and ethanol ranging between 20 and 25% of anhydrous ethanol), ethanol and flex fuel.." (Page 5, L161-162).

**Referee Comment: 153 How reasonable an assumption are emission factors measured in São Paulo? Do they depend on meteorological conditions?**

Author's Response: It not depend on meteorological conditions. However, it depends on some factors such as different vehicle types/fuel as specified in the manuscript. We decided to assume the emission factors of São Paulo, because it is the only database available in Brazil.

**Referee Comment: 177 Please include some examples of "transport and communication areas"**

Author's Response: Done. We included some examples: "..due to the production concentrated mainly in transport and communication areas such as electronics, metal mechanical sectors and the production of motorcycles" (Page 5-6, L189-190).

**Referee Comment: 192-196 Please explain why you used EPA rather than CETESB TPP emission factors.**

Author's Response: One of the reasons is because that the EPA has a complete list of air pollutants emission factors. For example, the emission factor of CO is not available in CETESB database. In addition, the HCNM emission should be share-according classes of WRF-chem, and this information is available in EPA. We explained this choice in the manuscript. The following sentence we included after "US Environmental Protection Agency (EPA, 1998 and 2010)": "which has a complete estimation of emission factor for all air pollutants simulated" (Page 6, L206-207).

**Referee Comment: 225 How do the emissions estimated in the present study compare with previous studies (previously existing inventories for Manaus, global inventories)?**

Author's Response: Actually, Manaus does not have an official and public inventory of air pollutants emissions. In relation to global inventories, they underestimate the emission calculated of anthropogenic emission sectors in this work (mobile and stationary sources). For instance, in MACCity anthropogenic emissions inventory, the sums approximate emissions in the grid study were 0.011 Tg/year (against 0.068), 0.003 Tg/year (against 0.087), 0.002 Tg/year (against 0.073) for CO, $NO_x$ and $SO_2$, respectively (see Table S1 in supplementary material).

**Referee Comment: 255-269 Include numerical values indicating "good representation of the temporal evolution..", "diurnal peak and minimum night temperatures were weakly represented.", "relative humidity profiles show a good level of agreement..", and "IFAM and T3 sites have the greatest discrepancies among the results.", and "According to the mean bias.."**

Author's Response: Done. We included numerical values in the following sentences:

" good representation of the temporal evolution of the temperature (e.g. r = 0.91 for T1)" (Page 8, L270);

", diurnal peak (e.g. 33.5 °C average observed against 30.3 °C simulated at 14:00 LT) and minimum night (e.g. 24.8 °C average observed against 26.3 °C simulated at 2:00 LT) temperatures were weakly represented "(Page 8, L274-276);

"The relative humidity profiles show a good level of agreement of the simulation to the average values of Colégio Militar (r = 0.89 and RMSE = 5.1) and T1 (r=0.88 and RMSE = 4.8) stations." (Page 8, L277-279);

"However, IFAM and T3 sites have the greatest discrepancies among the results, with MB values equal to -8.3 and -10.1, respectively" (Page 8, L279-280).

"According to the mean bias, it has been observed that the simulation, in general, underestimates the majority of observed values, mainly for temperature (T1, MB = -1.5) and relative humidity (IFAM, and T3). (Page 8, L287-288)".

**Referee Comment: 276-277 Provide values for MNBE**

Author's Response: Done. We included the values of MNBE in the following sentences: "..values observed (MNBE = 50.8).." (Page 8, L292-293)"; "...simulation period (MNBE equal a -29.5 and -35.4 for $NO_x$ and CO, respectively." (Page 8, L294)".

**Referee Comment: 280 What "fire outbreaks"? Do you mean that there was fire influence during this period and fire emissions were not included in the simulations? On lines 93-95 you state "Another important factor when choosing the period of study was the low incidence of biomass burning wildfires in the rainy season."**

Author's Response: One of the criteria of study period was the days that had a low incidence of fire outbreaks (see Figure S2 below). However, four fire outbreaks were occurred in the grid simulated, that may have influenced in the peak values of CO concentration. We included a brief comment in text "…that the fire outbreaks were not considered (four spots of fire occurred during the period inside the grid, although a low incidence of fire outbreaks was the criteria used for chosen the simulation period (Figure S2), which could have…" (Page 8, L298-299).

[Figure]

**Satellite**
- TERRA_M-T
- TERRA_M-M
- NPP
- NOAA-19D
- NOAA-18D
- NOAA-18
- NOAA-16
- NOAA-15D
- NOAA-15
- METEOSAT
- GOES-13
- AQUA_M-T
- AQUA_M-M

**Figure S2.** Map of fire outbreaks in Brazil and inside the studied grid during the simulation period.

**Referee Comment: 288-289 Would you expect a similar linear result if you varied mobile and stationary sources independently?**

Author's Response: Thanks for question. We clarify this sentence in the text, because the linear answer depends on pollutant analyzed and the atmospheric chemical and physical conditions such water, radiation, chemical regime (VOC/NOx), etc. For example, ozone not answer linearly to emission variation of VOC and NOx (Atkinson, 2000; Martins and Andrade, 2008). We changed the sentence: "Overall, it was observed that the model responds linearly to variations performed" to "Overall, it was observed that the model is sensible to the variations performed and for these pollutants the response is linear to it. However, O3 changes are not linear to emission variations of VOC and NOx (Atkinson, 2000; Martins and Andrade, 2008)" (Page 9, L307-310).

Atkinson, R. Atmospheric chemistry of VOCs and NO*x*. Atmospheric Environment Volume 34, Issues 12–14, 2000, Pages 2063–2101 http://dx.doi.org/10.1016/S1352-2310(99)00460-4

Martins, L.D., Andrade, M.F. Ozone Formation Potentials of Volatile Organic Compounds and Ozone Sensitivity to Their Emission in the Megacity of São Paulo, Brazil. Water Air Soil Pollut (2008) 195:201–213. DOI 10.1007/s11270-008-9740-x

**Referee Comment: 290-294 Is the range of 84-207 ppb calculated at the same altitude the aircraft sampled at, or in the first model level, where higher CO values would be expected?**

Author's Response: We calculated in the first level of the model. The highest concentrations are expected inside boundary layer.

**Referee Comment: 319 Do peak values have disproportionately higher impacts on public health? That is, are there threshold values of PM$_{2.5}$, NO$_x$, O$_3$ for health and ecosystem adverse impacts?**

Author's Response: Yes, in terms of health we have effects of short-time (short time exposure) and long-time exposition. In addition, the intake dose (relation between concentration and exposure time) is an important parameter to health effects. For example, ozone, which is a pollutant with short-time life is important to be analyzed the peak and average concentrations.

We showed in the temporal evaluation of NO$_x$ (see Figure 7 in the manuscript) that the peak values in the baseline scenario was more than 400 ppb. Moreover, in the future scenario these values exceed 600 ppb. These values are greater than those recommended, for example, by World Health Organization (WHO, 2005). Therefore, could have adverse impacts on health and ecosystem. The Brazilian and WHO standards are presented below:

| | Exposure time | PM$_{2.5}$ | PM$_{10}$ | TSP | SO$_2$ | NO$_2$ | Ozone | CO | Smoke |
|---|---|---|---|---|---|---|---|---|---|
| Brazilian standard (CONAMA no. 03/90) | Short-term | - | 150[b] (24 h) | 240[b] (24 h) | 365[b] (24 h) | 320 (1 h) | 160[b] (1 h) | 10,000[b] (8 h) 40,000[b] (1 h) | 150[b] (24 h) |
| | Long-term | - | 50 (year[c]) | 80 (year[d]) | 80 (year[c]) | 100 (year[c]) | - | - | 60 (year[c]) |
| WHO guidelines | Short-term | 25 (24 h) | 50 (24 h) | - | 20 (24 h) 500 (10 min) | 200 (1 h) | 100 (8 h) | 10,000 (8 h) 30,000 (1 h) | - |
| | Long-term | 10 (year[c]) | 20 (year[c]) | - | - | 40 (year[c]) | - | - | - |

TSP, Total Suspended Particles.

[a]All values expressed as µg.m$^{-3}$, except where otherwise indicated; [b]not to be exceeded more than once per year; [c]arithmetic annual mean; [d]geometric annual mean;

WHO. WHO Air quality guidelines global update – Report on a working group meeting 2005. Bonn: WHO, 2005.

**Referee Comment: 323-324 Unclear. Do you mean "In this sense, their impacts on peak average values should be evaluated."**

Author's Response: The authors clarify the sentence in the manuscript: "In this sense, their impacts on peak average values as in the spatial average should be evaluated" (Page 9, L345-346).

**Referee Comment: 345 Why was the contour of PM$_{2.5}$ ≥ 5 µg m$^{-3}$ chosen? Is it associated with some threshold of human health impact?**

Author's Response: We defined the value of the contour line of PM$_{2.5}$ according to the value calculated of 10% of contour line of SPTA (peak value) of future scenario.

**Referee Comment: 376-377 I recommend mentioning again that mobile/stationary emissions partitioning is important for designing regulations**

Author's Response: Thanks for the suggestion. We mentioned this information in the following sentence: "In this sense, to support new designing regulations, it is important to investigate the relative contribution of the mobile and stationary sources as well the area impacted by urban pollution plume (Page 11, L401-402)".

**Referee Comment: 379 How would you expect the results to change in the presence of biomass burning emissions?**

Author's Response: We will be expected that the peak values of some air pollutants will be changed and increase, mainly CO and particulate matter concentrations.

**Referee Comment: 422 Mention numbers again, e.g. "In all cases, an increase in air pollution concentrations (X%-X%) could be observed."**

Author's Response: Done. "In all cases, the increase in air pollution concentrations (e.g. 2020 scenario showed an increase of 12% in CO concentration) could be observed if the current conditions of the energy matrix were maintained." (Page 12, L446-448).

**Technical Comments**

**Referee Comment: 11 Insert "on air pollution" after "impact of the emissions from mobile and stationary sources."**

Author's Response: Done.

**Referee Comment: 14 Remove "Results show that"**

Author's Response: Done.

**Referee Comment: 16 Remove "ones"; I recommend replacing all instances of "futuristic scenario" with "future scenario."**

Author's Response: Done. We thank for the suggestion and it was replaced "futuristic scenario" to "future scenario" throughout the text.

**Referee Comment: 17-18 Replace two instances of "has shown" with "showed"**

Author's Response: Done.

**Referee Comment: 19 Insert "transported" after "predominantly."**

Author's Response: Done.

**Referee Comment: 31 Alter "oxide nitrogen" to "nitrogen oxides"**

Author's Response: Done.

**Referee Comment: 33-34 Alter "technologies, which guarantee" to "technologies that guarantee"**

Author's Response: Done.

**Referee Comment: 37 Alter "the complex combination of the latter" to "their complex combination"**

Author's Response: Done.

**Referee Comment: 38 Alter "the transportation and the combination of atmospheric pollutants" to "the transport and interactions of atmospheric pollutants"**

Author's Response: Done.

**Referee Comment: 41-42 Alter "aspects of the pollution impact. For instance, pollution episodes." to "aspects of the pollution impact, such as pollution episodes."**

Author's Response: Done.

**Referee Comment: 43 Alter "transportation" to "transport"**

Author's Response: Done.

**Referee Comment: 44 Alter "effects on" to "effects of"**

Author's Response: Done.

**Referee Comment: 46 Alter "Such studies are not capable of investigating the impact of isolated urban plumes." to "Such studies did not investigate the impact of isolated urban plumes."**

Author's Response: Done.

**Referee Comment: 47 Alter "in the last decades" to "in recent decades"**

Author's Response: Done.

**Referee Comment: 58 Alter "Bela et al., 2014" to "Bela et al., 2015.";  Alter "Therefore, such studies" to "These studies".**

Author's Response: Done.

**Referee Comment: 62 Insert "pollutant concentrations above" before "the preserved forest region"**

Author's Response: Done.

**Referee Comment: 69 Alter "Participation of mobile and stationary sources in.." to "Participation of mobile and stationary anthropogenic sources and biogenic sources in.."**

Author's Response: Done.

**Referee Comment: 76 Alter "scope" to "region"**

Author's Response: Done.

**Referee Comment: 77 and 215 Remove "$\lambda =$" and "$\varphi =$"**

Author's Response: Done.

**Referee Comment: 81 Alter "represents" to "is part of"**

Author's Response: Done.

**Referee Comment: 101-102 Move "simultaneously" to before "predicts"**

Author's Response: Done.

**Referee Comment: 113 Delete "side"**

Author's Response: Done.

**Referee Comment: 118 Insert "reactions" after "photolysis"**

Author's Response: Done.

**Referee Comment: 121 Insert "masl" after "height"**
Author Response: Done

**Referee Comment: 150 Alter "grid study" to "study grid"**

Author's Response: Done.

**Referee Comment: 200 Alter "Northern region" to "North region"**

Author's Response: Done.

**Referee Comment: 202-203 Alter two instances of "admitted" to "assumed"**

Author's Response: Done.

**Referee Comment: 207 Alter "greater" to "greatest"**

Author's Response: Done.

**Referee Comment: 214 Alter "northern region" to "North region"**

Author's Response: Done.

**Referee Comment: 273 Remove "In relation to air pollutants"**

Author's Response: Done.

**Referee Comment: 276-277 Alter "with CO and NOx, the tendency is the opposite. They underestimated.." to "with CO and NOx, the simulations underestimated.."**

Author's Response: Done.

**Referee Comment: 288-289 Move "Temporal evolution of $PM_{10}$ and $NO_x$ concentrations is shown in Figure S1." before "Overall, it was observed that the model responds linearly to variations performed."**

Author's Response: Done.

**Referee Comment:** 300-303 Insert "downwind of Manaus" after "O3 mixing ratios."

Author's Response: Done.

**Referee Comment:** 313 Insert "significant" before "effects"

Author's Response: Done.

**Referee Comment:** 314 Alter "in this work" to "in the present study"

Author's Response: Done.

**Referee Comment:** 323 Alter "its impact" to "their impact"

Author's Response: Done.

**Referee Comment:** 328-329 Alter "in each extremity. The values recommended by Skamarock et al. (2008) were used in order to reduce the effect of lateral boundary conditions." to "in each extremity, which were the values recommended by Skamarock et al. (2008) to reduce the effect of lateral boundary conditions."

Author's Response: Done.

**Referee Comment:** 332 Insert "spatial" before "average"

Author's Response: Done.

**Referee Comment:** 335 Alter "parameters applied" to parameters considered"

Author's Response: Done.

**Referee Comment:** 345 Alter In order to analyze the plume of Manaus city, the spatial distributions of pollutants evaluated have been performed." to "In order to analyze the spatial extent and location of impact of the plume of Manaus city, the spatial distributions of pollutants have been evaluated."

Author's Response: Done.

**Referee Comment:** 347-348 Alter "The impact is both on the average values and the peak values." to "The impact is both on the spatial average and spatial peak values."

Author's Response: Done.

**Referee Comment:** 375 Insert "not present in global inventories" after "daily cycle of emissions

Author's Response: Done.

**Referee Comment:** 427 "Network for Climate"

Author's Response: Done.

**Referee Comment:** Table 1 - Caption - Alter "Physical parameterization used for this study." to "WRF-Chem physics parameterizations used for this study."; Alter "Soil-Land Parameterization" to "Soil-Land"

Author's Response: Done.

**Referee Comment::** Table 3 - Caption – Alter "per" to "by"

Author's Response: Done.

**Referee Comment:** Table 4 - Alter four instances of "natural" to "biogenic"

Author's Response: Done.

**Referee Comment:** Table 9 - Insert units in parentheses after "Area"

Author's Response: Done. We included "($km^2$)" after "Area".

**Referee Comment:** Figure 2 - Could you combine Fig. 2 with Fig. 1 to save space?

Author Response: We tried to combine Fig. 1 and 2. However, due to large number of thermal power plants in Manaus city, the Figure compromised the visualization of meteorological and air quality stations. Therefore, we decided to leave the both figures in the manuscript.

**Referee Comment:** Figure 5 - Caption – format subscripts for $PM_{2.5}$ and $NO_x$

Author's Response: Done.

---

## Author Comment (AC2)

Dear Dr. Hugo Abi Karam.

We would like to thank you for your insightful comments that enabled us to improve the quality of our manuscript. Detailed responses to your concerns are outlined below.

**Referee Comment: Show each focus of heat and smoke on the map of South America during the days of the experiment, obtained from satellite imagery. One might ask what can be stated about the pollutant plume emitted from each heat source, especially around Manaus, with trajectories within the PBL. If the heat sources were many, or a focus is highlighted from the others by the intensity, what would be the axis of the dispersing smoke plume in the vicinity of Manaus? In addition, what is the concentration of transient pollutants on the area of interest? It is not enough to indicate in the text the absence of ´ heat / smoke focus in the area because a distant focus may be the source of a pollutant feather that propagates through long trajectories to the vicinity of Manaus. Could you verify this?**

Author's Response: We included a map of fire outbreaks in Brazil and inside the studied grid during the simulation period in the electronic supplementary material (see Figure S2 below). In addition, we included a brief comment in the text "…that the fire outbreaks were not considered (four spots of fire occurred during the period inside the grid, although a low incidence of fire outbreaks was the criteria used for chosen the simulation period (Figure S2), which could have…" (Page 8, L298-299).

[Figure]

**Figure S2**. Map of fire outbreaks in Brazil and inside the studied grid during the simulation period.

**Referee Comment:** **Show if possible the spatial distribution of pollutant emission rates on the surface of the domain, graphically presenting the result of the inventory prepared and used. In addition, show corresponding two-dimensional figures. For the emission of $CO_2$ and other pollutants of daytime variation, present maps of the emission rate every 3 hours over the 24 hour cycle.**

Author's Response: We understand that the inclusion of such figures would provide a good idea of how the emissions are distributed in the spatial domain. However, we decide to provide this information in a different way in the Supplementary Material and we hope it can satisfy your request in some way. We included Table S1, which contains the emissions for all locations in our grid domain. The emissions are provided in terms of mass of pollutant per year. Diurnal emissions are just considered by dividing this number by 365 and distributed following the diurnal cycle depending on the source type, as we explain in the sequence. Additionally, Figure 3 of the manuscript shows a partition of the emissions for $NO_x$ and $SO_2$ for the different sources.

In relation to the diurnal cycle of the emissions, it is based on the work of Andrade et al., 2015 (Figure 7 of their work). The cycle considers the two hush ours for vehicular emissions and also the different pattern for Heavy Duty Vehicles (HDV). For fixed sources, as the thermal power plants, emissions are constant in time.

**Referee Comment: Characterize the synoptic and mesoscale conditions present during the days of the numerical investigation. Satellite images are available? Look for the channels whose composition matches the image of air masses moving in the domain, thus doing to highlight the aerosol plumes if possible. Can this be done?**

Author's Response: Done. We included GOES -13 water vapor images in the electronic supplementary material (see Figure S4 below).

[Figure]

[Figure]

**Figure S4.** GOES -13 water vapor images every 6 hours starting on March 17 at 04Z.

**Referee Comment: Spatial fields can be presented to characterize the synoptic condition: current lines and advection of equivalent potential temperature.**

Author's Response: Done. We included spatial fields of streamlines and advection of equivalent potential temperature in 925 hPa of analysis data from the Global Model Data Assimilation System (GDAS) in the electronic supplementary material (see Figure S1 below). The following sentence was mentioned in the text "Figure S1 of the Supplementary Material shows a time evolution of streamlines and the Potential Temperature advection, where these atmospheric conditions can be seen" (Page 3, L97-99).

[Figure]

**Figure S1.** Spatial fields of streamlines and advection of equivalent potential temperature in 925 hPa. The black contour line shows the delimitation of the city of Manaus.

---

## Author Response (AR2)

Dear Editor,

Point-by-point responses to all report comments are listed below.

**Referee report #1**

I recommend that the paper be accepted for publication in ACP after the following minor revisions:

**Comment:** 60 After "which require the preparation of an inventory for mobile and stationary sources" insert ", since no official and public emissions inventory is available for Manaus."

**Response:** Done.

**Comment:** 160 After "Brito et al., 2013)" insert ", which are the only vehicle emission factor measurements available in Brazil."

**Response:** Done.

**Comment:** 234 Include a reference for the MACCity data, and after "shown in Table S1 in the electronic supplementary material." insert "These results demonstrate that global inventories underestimate mobile and stationary sources in Manaus. For instance, in the MACCity anthropogenic emissions inventory, the approximate sums of emissions in the study grid were 0.011 Tg/year (versus 0.068 in this study), 0.003 Tg/year (versus 0.087 in this study), 0.002 Tg/year (versus 0.073 in this study) for CO, NOx and SO2, respectively."

**Response:** Done.

**Comment:** 302 Alter "sensible" to "sensitive"

**Response:** Done.

**Comment:** 303 Alter "for these pollutants the response is linear to it" to "for $PM_{10}$ and NOx the responses to the variations are linear"

**Response:** Done.

**Comment:** 309 Please explain/clarify what you mean by "most differences".

**Response:** We changed the sentence to make it clearer: from "Considering the scenario C0, representing the emission inventory of current conditions for the region, the values found in this study stayed in the 84-207 ppb range, with most differences being observed in the first levels of the model" to "Considering the scenario C0, representing the emission inventory of current conditions for the region, the values found in this study stayed in the 84-207 ppb range, where the most variations in the concentration were observed in the lower vertical levels of the model, as expected".

**Comment:** 369 After greater than or equal to 5 ug m$^{-3}$," insert " which is the value of 10% of the SPTA (peak value) of the future scenario,"

**Response:** Done.